# Effect of a brief psychological intervention for common mental disorders on HIV viral suppression: A non-randomised controlled study of the Friendship Bench in Zimbabwe

**Victoria Simms**[1]☯*, **Melanie A. Abas**[2]☯*, **Monika Müller**[2,3], **Epiphania Munetsi**[4], **Lloyd Dzapasi**[4], **Helen A. Weiss**[1], **Dixon Chibanda**[4,5,6]

**1** MRC International Statistics and Epidemiology Group, London School of Hygiene & Tropical Medicine, London, United Kingdom, **2** Centre for Global Mental Health, Institute of Psychiatry, Psychology and Neuroscience, King's College London, London, United Kingdom, **3** University Hospital of Psychiatry Bern, Bern, Switzerland, **4** Friendship Bench, Harare, Zimbabwe, **5** Department of Psychiatry, University of Zimbabwe, Harare, Zimbabwe, **6** Centre For Global Mental Health, London School of Hygiene and Tropical Medicine, London, United Kingdom

☯ These authors contributed equally to this work.
* victoria.simms@lshtm.ac.uk (VS); melanie.abas@kcl.ac.uk (MAA)

**Data Availability Statement:** The dataset has been submitted to the curated LSHTM DataCompass

## Abstract

### Background

For people living with co-morbid HIV and common mental disorders (CMD), it is not known whether a brief psychological intervention for CMD can improve HIV viral suppression.

### Methods

We conducted a prospective cohort study in eight primary care clinics in Harare, Zimbabwe, enrolling adults with co-morbid HIV and CMD. Six clinics provided the Friendship Bench (FB), a brief psychological intervention for CMD based on problem-solving therapy, delivered by lay counsellors. Two clinics provided enhanced usual care (EUC). The primary outcome was viral non-suppression after six months (viral load ≥400 copies/mL). Data were analysed using a difference-in-difference approach with linear regression of cluster-level proportions, adjusted for baseline viral non-suppression (aDiD). The secondary outcome was presence of CMD measured by the Shona Symptom Questionnaire.

### Results

In FB clinics, 407/500 (81.4%) participants had viral load results at baseline and endline: 58 (14.3%) had viral non-suppression at baseline and 41 (10.1%) at endline. In EUC clinics, 172/200 (86.0%) had viral load results at baseline and endline: 22 (12.8%) were non-suppressed at baseline and 26 (15.1%) at endline (aDiD = -7.3%; 95%CI 14.7% to -0.01%; p = 0.05). Of the 499 participants virally suppressed at baseline, the FB group had lower prevalence of non-suppression at endline compared to the EUC group (2.9% vs 9.3%; p = 0.002). There was no evidence of a difference in endline viral non-

repository and will be available for download
(https://doi.org/10.17037/DATA.00003219).

**Funding:** This study was funded through a grant
from the National Institute for Mental Health
(NIMH), and The U.S. President's Emergency Plan
for AIDS Relief (PEPFAR) awarded to DC. EM and
LD received part-time salary from this PEPFAR/
NIMH grant. VS and HAW are supported by the UK
Medical Research Council (MRC) and the UK
Department for International Development (DFID)
under the MRC/DFID Concordat agreement which
is also part of the EDCTP2 programme supported
by the European Union (MR/R010161/1). The
funders had no role in study design, data collection
and analysis, decision to publish, or preparation of
the manuscript.

**Competing interests:** The authors have declared
that no competing interests exist.

suppression by group among the 80 participants with non-suppression at baseline
(53.5% vs 54.6%; p = 0.93). The FB group was less likely to screen positive for CMD at
endline than the EUC group (aDiD = -21.6%; 95%CI -36.5% to -6.7%; p = 0.008).

## Conclusion

People living with co-morbid HIV and CMD may benefit from receiving a low-cost mental
health intervention to enhance viral suppression, especially if they are already virally sup-
pressed. Research is needed to understand if additional adherence counselling could fur-
ther improve viral suppression.

## Introduction

Sustained viral suppression is a key goal of care for people living with HIV (PLWH)
because it halts progression to AIDS, enables near-normal life expectancy and prevents
transmission to sexual partners [1]. Antiretroviral therapy (ART) taken as prescribed is the
key driver of viral suppression [2], with a recent systematic review finding that a diverse
range of interventions are helpful in improving ART adherence [3]. Common mental dis-
orders (CMD), including depression and anxiety increase risk of poor adherence to ART
globally [4, 5], including in Zimbabwe [6], contributing to viral non-suppression and
mortality [7, 8]. CMD are very frequent in PLWH living in countries in sub-Saharan Africa
[9, 10]. A recent global systematic review reported 28–62% of PLWH have mental health
symptoms, especially depression, anxiety and suicidal ideation [11]. The WHO and
UNAIDS have recently emphasised the importance of addressing mental health in PLWH
[12].

A recent systematic review of randomised controlled trials (RCT) found that task-shifted
psychological therapies are effective for mental disorders in PLWH in low- and middle-
income countries (LMICs) [13]. This review did not identify any RCTs from LMICs evaluat-
ing the effectiveness of CMD treatment on ART adherence and/or viral suppression in
PLWH [14]. A widely-cited systematic review and meta-analysis, based almost exclusively on
studies from high-income countries, suggested that treatment of depression improves ART
adherence (standardized relative risk 1.35, 95% CI 1.13–1.60) [15]. However, many studies
included in this review were prone to recall bias or described interventions which included
adherence counselling as well as depression treatment. There remains an evidence gap on
whether treatment of depression alone, without additional adherence support, improves
viral suppression.

The Friendship Bench (FB) is a brief psychological intervention for depression and
other CMD based on problem-solving therapy and simple behavioural activation [16]. The
FB was developed in Zimbabwe and is delivered by trained lay counsellors in primary care
[17]. A cluster-randomised controlled trial showed that FB was effective at improving
symptoms of CMD and of depression in adults with and without HIV who accessed pri-
mary care [18].

The aim of this comparative cohort study was to evaluate the effect of FB on HIV viral sup-
pression and mental health outcomes among PLWH with co-morbid CMD in Harare, Zimba-
bwe. Our hypothesis was that receiving the Friendship Bench, compared to usual care, is
associated with viral suppression at 6 months follow-up.

## Materials and methods

### Study design and setting

The study was a multicentre prospective cohort study of PLWH with co-morbid CMD, who attended one of eight primary care clinics in high- or middle-density suburbs of Harare. When the study began (August 2017), scale-up of the FB was taking place in primary care clinics in Harare. Six FB clinics were purposively selected for their large patient volume (>300/day). Of the primary care clinics that did not yet offer FB, the two largest in terms of patient volume were selected as enhanced usual care (EUC) clinics. All clinics provided HIV care and other services including acute primary care, chronic disease outpatients, family planning, and maternity at the primary care level. None routinely provided specialist mental health care. They were typically staffed by up to 50 health care professionals with an average of 20 nurses and two physicians.

### Recruitment

We used systematic sampling to select every third patient at each clinic (based on their allocated queue number) to screen for eligibility. PLWH aged 18 years or above were eligible if they had been on ART for at least three months, attended HIV care at the clinic in person, lived in the clinic catchment area and had co-morbid CMD diagnosed using the Shona Symptom Questionnaire (SSQ-14≥9) [19]. We excluded participants if they were receiving mental health care in a psychiatric unit, presented with suicidal intent, psychotic symptoms, intoxication, or dementia. We defined participants as 'red flag' cases if they had a SSQ-14≥11 or screened positive for either suicidal ideation ("Did you sometimes feel like taking your own life") or for psychotic symptoms ("Did you see or hear things which others could not see or hear"). These participants were subsequently assessed by a senior lay counselor with supervisor capacity and re-screened using the Patient Health Questionnaire (PHQ-9) for depression. If participants were deemed by the senior counselor and the study psychologist to have suicidal intent, or if psychotic symptoms were confirmed, they were not eligible for the study and were referred for specialist care at a tertiary care facility. Those who were eligible for the study were evaluated by the senior counsellor and referred back to the FB.

Eligible participants were asked for written informed consent to participate in the study which included access to their routine medical records. Participants received US $3 for each study related visit to cover transport costs. Ethical approval was granted by the Medical Research Council of Zimbabwe (MRCZ A/2130) and the ethics committees of the London School of Hygiene & Tropical Medicine (Ref 11759). The study observed the STROBE guideline for reporting on observational studies [20].

### Data collection at baseline and endline

Eight research assistants interviewed participants at baseline and at endline (six months post-enrolment) using a tablet-based questionnaire to collect sociodemographic and mental health outcomes. The questionnaire was developed based on similar studies previously conducted in this setting [18, 19]. Participants who did not present for their endline visit were contacted by phone or through significant others using contact data collected at baseline.

Socio-demographic characteristics included gender, age, marital status, education, income, type of housing, number of persons sleeping in one bedroom including index person, and current alcohol use. Overcrowding was defined as the index person sleeping with three or more additional persons in the room. We obtained date of ART initiation and viral load at baseline from clinical records, accepting viral loads completed up to two weeks post-enrolment. At

endline we collected viral load specifically for the study. We used the following validated questionnaires for mental health assessment:

**The Shona Symptom Questionnaire (SSQ-14) [19, 21].** The SSQ-14 is a measure of CMD developed and validated in Zimbabwe which includes locally accepted idioms of distress. It consists of 14 binary items and is scored from 0–14, with higher scores indicating more symptoms. A cutoff of ≥9 provides 88% sensitivity and 76% specificity among PLWH to diagnose CMD against a standardised diagnosis of depression and/or general anxiety disorder [19], with good internal reliability (Cronbach α = 0.74).

**The Patient Health Questionnaire (PHQ-9) [19, 22].** The PHQ-9 measures depression based on the Diagnostic and Statistical Manual for Mental Disorders (DSM)-IV and DSM-V criteria. It consists of nine items with 4-point Likert responses and scores from 0–27, with higher scores indicating worse symptoms. The Shona language version was validated in primary care in Zimbabwe against a diagnosis of depression. An optimal cut-off of ≥11 provided a sensitivity of 85% and specificity of 69%, with high internal reliability (Cronbach α = 0.86) to detect depression.

**The Generalized Anxiety Disorder questionnaire (GAD-7) [19, 22].** The GAD-7 is a screening tool for general anxiety disorder which consists of seven items with four-point Likert responses and scores from 0 to 21, with higher scores suggesting worse symptoms. The Shona language version was validated in a Zimbabwean primary care context against a diagnosis of GAD. Its optimal cut-off of ≥10 showed a sensitivity of 89% and specificity of 73% with high internal reliability (Cronbach α = 0.87). We defined the severity of anxiety at baseline according to the established GAD-7 categories (0–4 no, 5–9 mild, 10–14 moderate and 15–21 severe anxiety).

**The WHO Disability Assessment Schedule (WHO-DAS 2) [19].** The WHO-DAS 2 is a 12-item score for disability which ranges from 0–48, with higher scores indicating higher disability.

## Friendship bench intervention

The FB intervention consists of six sessions of culturally adapted problem-solving therapy and simple behavioural activation for depression delivered by a trained lay counsellor on a bench in a discreet area outside the clinic [16, 17]. The first session was delivered on the day of enrolment with subsequent weekly sessions. If a participant missed a session, the counsellor got in touch by phone or SMS to encourage adherence to problem-solving therapy and re-schedule the next session. Patients were also invited to a peer-led support group with an income generation component, known as Circle Kubatana Tose. Twelve trained lay health counsellors employed by the City of Harare Health Department delivered the FB intervention, supervised on a two-weekly basis by a mental health care professional from the FB team. All counsellors were experienced in delivering the intervention and had treated patients in the previous FB clinical trial [18] The initial training lasted nine days and included knowledge on common mental disorders, counselling skills, problem-solving therapy, and self-care. All lay counsellors attended a group discussion where lessons from the earlier trial were shared, including how they addressed depression in PLWH. A five-day refresher training on problem-solving therapy was then carried out with core modules on counselling PLWH and management of participants with 'red flags' to empower the task-shifting approach and limit referral to tertiary care.

## Enhanced usual care

The comparison group received EUC consisting of nurse-led psychoeducation and assessment for any need for further mental health care based on the Mental Health Gap Action Program (mhGAP) of the WHO [23]. Further mental health care included the prescription of

antidepressant medication by the clinic nurse and strengthened referral to existing mental health care services for participants showing a 'red flag' as defined above. These were also available in the FB group.

### HIV care

All study participants received standard HIV care. In case of viral non-suppression nurse-led enhanced adherence counselling was conducted in accordance with Ministry of Health recommendations [24]. This consisted of three sessions on the benefits of ART and an in-depth discussion on reasons for non-adherence with the option to refer to a physician.

### Outcomes

The primary outcome was the proportion of participants with viral non-suppression at six months follow-up, defined as a viral load ≥400 copies/mL or death. Secondary outcomes were mental health symptoms assessed using the SSQ-14, PHQ-9, GAD-7, and WHO-DAS. These were analysed as continuous scores and proportions screening positive as defined above (SSQ-14 ≥9, PHQ-9 ≥11, GAD-7 ≥10 and WHO-DAS ≥20).

### Statistical analysis

The study had 81% power to detect a difference in viral non-suppression at 6 months of 30% in the EUC clinics vs 18% in the FB clinics, based on recruiting 750 participants from ten FB clinics and 250 from three EUC clinics, with an ICC of 0.01. The ZIMPHIA report 2016 [25] found that 15% of adults in HIV care in Zimbabwe were non-suppressed; we estimated this would be twice as high among people with CMDs.

Analyses were conducted at cluster-level due to the small number of clusters [26]. The primary analysis was a cluster-level difference-in-difference approach assessed using linear regression of cluster-level proportions or means on FB vs EUC, adjusted for baseline viral non-suppression (aDiD). We used individual-level multivariable logistic regression to identify potential confounders (i.e. baseline variables that were independently associated with treatment group and with viral non-suppression at follow-up, adjusting for clinic using robust standard errors). We identified baseline variables independently associated with loss-to-follow-up following the same statistical approach. Potential confounders or variables associated with loss-to-follow-up were included in the initial aDiD model if they changed the effect estimate by more than 10%. The p-value was estimated assuming a t-distribution to account for the small degrees of freedom. The same approach was used for the mental health outcomes, adjusting for the respective baseline mental health value. For the primary outcome we also conducted an analysis stratified by baseline viral suppression.

We conducted two post-hoc sensitivity analyses of the primary outcome. First, we included only participants with less than eight months follow-up to account for differences in follow-up time between the treatment groups. Second, we excluded participants recruited in the last three months of the recruitment period. During this time, data on reasons for non-eligibility were not collected, which might have introduced a selection bias. All analyses were conducted in Stata version 17.0.

## Results

### Study flow

The original target was to enrol 75 participants per clinic in 10 FB clinics (N = 750) and 83 per clinic in three EUC clinics (N = 249). However, fewer PLWH were attending per day than

expected, especially in four FB clinics and one EUC clinic. A key reason for this was relatives and friends of PLWH coming to collect repeat prescriptions for them. Thus we reduced the sample size to six FB and two EUC clinics, with enrolment targets of 80–100 per FB clinic (total N = 500) and 100 per EUC clinic (total N = 200). Between August 2017 and July 2018, 2019 PLWH were screened in the six FB clinics, of whom 543 (26.9%) were eligible, and 500 (92.1%) were enrolled (Fig 1). In the two EUC clinics 817 PLWH were screened, of whom 212 (25.9%) were eligible, and 200 (94.3%) were enrolled.

The main reason for non-eligibility in both FB and EUC clinics was not having CMD (n = 1704). Even though 99 participants reported hallucinations and 144 reported suicidal ideation when screened for CMD at baseline, none of these patients were deemed to have suicidal intent or psychotic symptoms by the study psychologist and thus we did not exclude any participants due to psychiatric reasons. Data on reasons for exclusion was not available for participants screened in the last three months of recruitment (May-July 2018), hence they are listed as 'unknown' (2.5% of those screened in the FB clinics and 14.0% in the EUC clinics).

Overall, 579 participants (82.7%) had viral load test results at both baseline and endline (81.4% and 86.0% in the FB and EUC group, respectively). The most common reason for non-availability of baseline viral loads was collection outside the window period of two weeks post-enrolment. The median duration from enrolment to follow-up viral load test was 6.5 months (IQR 5.3–7.8) in the FB group and 8.1 months (IQR 6.8–11.7) in the EUC group (p<0.001). A total of 568 (81.1%) participants had mental health outcome data at follow-up (81.2% and 81.0% in the FB and EUC groups, respectively). The median duration of follow-up for the

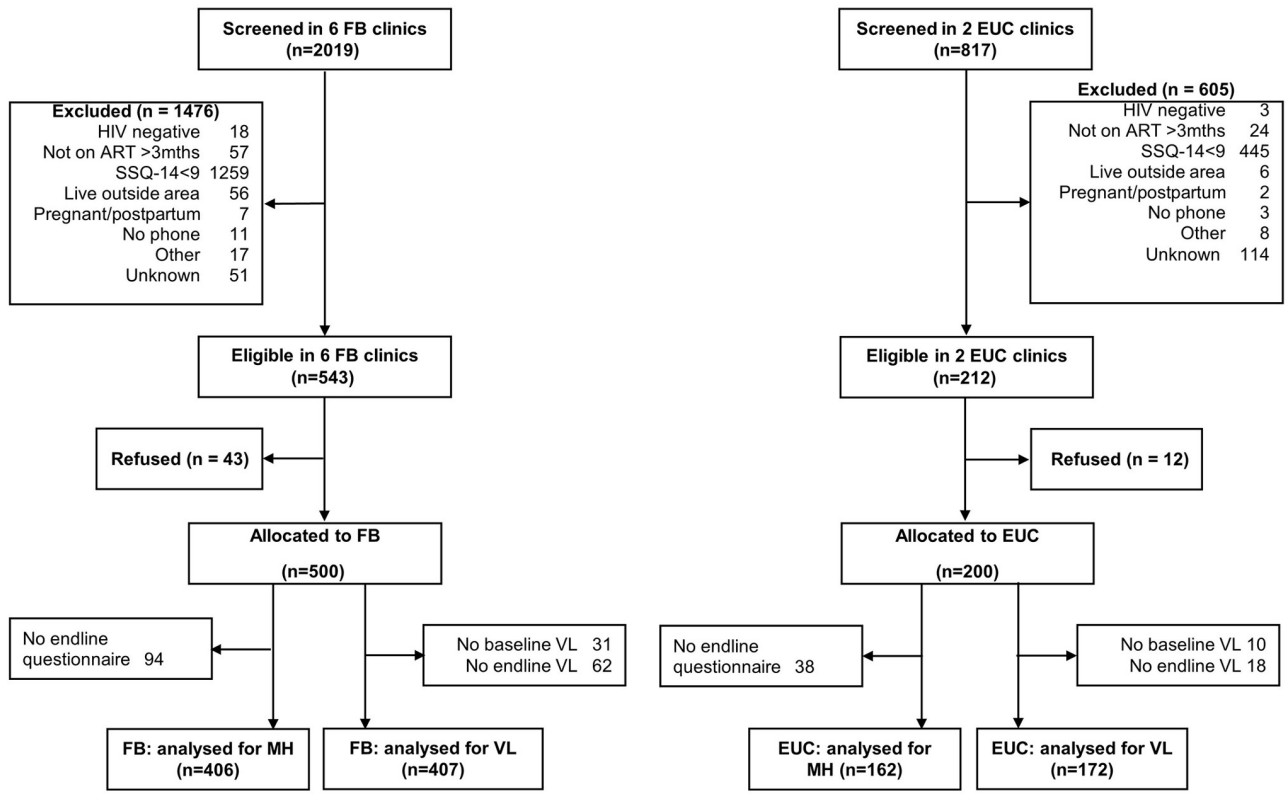

**Fig 1. Participants living with HIV and CMD recruited into the study between August 2017 and July 2018.**

mental health outcomes was 6.5 months (IQR 5.2–7.2) in the FB group and 9.5 months (IQR 7.0–12.8) in the EUC group (p<0.001).

## Study population characteristics

Table 1 shows baseline characteristics of the 579 participants with complete viral load data (i.e. at baseline and endline) by treatment group. The mean age at enrolment was 40.6 years (SD 9.9 years). Most participants (82.0%) were female. About half the participants were married and most had some secondary education (Table 1). Most (89.1%) participants stated that they had some income being either employed, supported by the family and/or partner or running their own business Most (89.6%) had at least one child and 37.3% were caring for other children than their own, including grandchildren, with a similar distribution in both groups. Of all participants, 13.6% lived in overcrowded conditions and 13.1% drank alcohol, but only five participants reported they consumed alcohol daily. At baseline, 80 (13.8%) participants had viral non-suppression (14.3% and 12.8% in the FB and EUC groups respectively). As shown in Table 1, median duration on ART in both groups was similar, as was severity of CMD. Prevalence of major depression in the whole sample (PHQ-9 ≥11) was 55.8% and of clinically relevant general anxiety disorder (GAD-7 ≥10) was 47.0%. Sixty-seven (11.6%) participants in the whole sample showed severe depression (PHQ-9 ≥20), and 132 (22.8%) severe generalised anxiety disorder (GAD-7 ≥15). Almost half of the participants registered a 'red flag' (50.1% and 45.4% in the FB and EUC groups respectively). All were assessed by senior counsellors but none were deemed in need of referral to tertiary care or started on antidepressants during the study period.

Age group and time since ART initiation were associated with treatment group (Table 1) and endline viral non-suppression (S1 Table) and were deemed potential confounders, but did not act as confounders as they changed the effect estimate by less than 10%. Viral load missingness was associated with marital status, income, alcohol consumption, length of time on ART, and having an SSQ red flag (S2 Table). Of these, marital status, income and years on ART were also associated with treatment group but did not act as confounders. Therefore, we did not adjust for any additional baseline characteristics in the final analysis.

## Association of the Friendship Bench with viral non-suppression

The prevalence of viral non-suppression at baseline and endline stratified by clinic is shown in Fig 2. All FB clinics showed a reduction in viral non-suppression between baseline and endline, whereas both EUC clinics showed an increase. Table 2 summarizes the proportion with viral non-suppression at endline and the results of the aDiD analysis. Overall, 10.1% (41/407) of the participants in FB group had viral non-suppression at endline as compared to 15.1% (26/172) at EUC group (aDiD = -7.3%; 95%CI -14.7% to -0.01%; p = 0.05). Among the 499 participants who were virally supressed at baseline, prevalence of viral non-suppression at follow-up was lower in the FB group than the EUC group. (2.9% vs 9.3%; DiD -6.5% (95% CI -10.5% to -2.4%), p = 0.002). Among the 80 participants with viral non-suppression at baseline, there was no evidence of a difference in viral non-suppression at endline (53.5% vs 54.5%; p = 0.93).

## Association of the Friendship Bench with mental health

Table 3 summarizes the associations of the FB with mental health. There was strong evidence that participants in the FB group had lower prevalence of CMD at endline (SSQ-14 ≥9) than those in the EUD group (aDID = -21.6%; 95%CI -36.5% to -6.7%; p = 0.008). There was a similar pattern for depression (PHQ-9 ≥11) and anxiety (GAD-7 ≥19), although the associations

**Table 1. Baseline characteristics of 579 participants with viral load testing at baseline and follow-up according to treatment group.**

| | | EUC n (%) or mean (SD) N = 172 | FB n (%) or mean (SD) N = 407 | p |
|---|---|---|---|---|
| **SOCIODEMOGRAPHIC CHARACTERISTICS** | | | | |
| **Gender** | Male | 45 (26.2%) | 59 (14.5%) | $X^2$ = 11.2, 0.001 |
| | Female | 127 (73.8%) | 348 (85.5%) | |
| **Age group** | 18–29 | 19 (11.1%) | 53 (13.1%) | $X^2$ = 3.2, p = 0.52 |
| | 30–39 | 61 (35.5%) | 139 (34.2%) | |
| | 40–49 | 68 (39.5%) | 138 (33.9%) | |
| | 50–59 | 19 (11.1%) | 60 (14.7%) | |
| | 60–72 | 5 (2.9%) | 17 (4.2%) | |
| **Marital status** | Married | 94 (54.7%) | 185 (45.5%) | $X^2$ = 11.8, p = 0.003 |
| | Single | 51 (29.7%) | 103 (25.3%) | |
| | Widowed | 27 (15.7%) | 119 (29.2%) | |
| **Highest level of education achieved** | Incomplete primary | 8 (4.7%) | 23 (5.7%) | $X^2$ = 9.4, p = 0.05 |
| | Complete primary | 38 (22.1%) | 58 (14.3%) | |
| | Incomplete secondary | 57 (33.1%) | 119 (29.2%) | |
| | Complete secondary | 61 (35.5%) | 192 (47.2%) | |
| | Tertiary | 8 (4.7%) | 15 (3.7%) | |
| **Income (N = 698)** | No | 10 (5.8%) | 53 (13.1%) | $X^2$ = 6.6, p = 0.01 |
| | Yes | 162 (94.2%) | 352 (86.9%) | |
| **Living in a house (N = 574)** | No | 26 (15.4%) | 69 (17.0%) | $X^2$ = 0.2, p = 0.63 |
| | Yes | 143 (84.6%) | 336 (83.0%) | |
| **Overcrowding$** | No | 148 (86.0%) | 352 (86.5%) | $X^2$ = 0.02, p = 0.89 |
| | Yes | 24 (14.0%) | 55 (13.5%) | |
| **Drink alcohol** | No | 145 (84.3%) | 358 (88.0%) | $X^2$ = 1.4, p = 0.23 |
| | Yes | 27 (15.7%) | 49 (12.0%) | |
| **HIV RELATED CHARACTERISTICS** | | | | |
| **Baseline HIV Viral load** | <400 copies/mL | 150 (87.2%) | 349 (85.8%) | $X^2$ = 0.2, p = 0.64 |
| | ≥400 copies/mL | 22 (12.8%) | 58 (14.3%) | |
| **Years since ART initiation N = 547** | 0 | 27 (15.7%) | 64 (15.9%) | $X^2$ = 8.3, p = 0.14 |
| | 1 | 23 (13.4%) | 46 (11.4%) | |
| | 2 | 10 (5.8%) | 51 (12.7%) | |
| | 3 | 20 (11.6%) | 30 (7.5%) | |
| | 4 | 13 (7.6%) | 33 (8.2%) | |
| | ≥ 5 | 79 (45.9%) | 178 (44.3%) | |
| **MENTAL HEALTH CHARACTERISTICS** | | | | |
| **Depression severity (PHQ-9 score)** | | 11.3 (5.1) | 12.1 (6.1) | t = -1.53, p = 0.13 |
| **Major depression (PHQ-9 ≥11)** | No | 77 (44.7%) | 179 (44.0%) | $X^2$ = 0.03, p = 0.86 |
| | Yes | 95 (55.2%) | 228 (56.0%) | |
| **Anxiety severity (GAD-7 score)** | | 9.2 (4.9) | 10.4 (5.4) | t = -2.58, p = 0.01 |
| **Anxiety disorder (GAD-7 ≥10)** | No | 97 (56.4%) | 210 (51.6%) | $X^2$ = 1.1, p = 0.29 |
| | Yes | 75 (43.6%) | 197 (48.4%) | |
| **Risk identified \*** | No red flag | 94 (54.7%) | 203 (49.9%) | $X^2$ = 1.1, p = 0.29 |
| | Red flag | 78 (45.4%) | 204 (50.1%) | |

*(Continued)*

**Table 1.** (Continued)

| | | EUC n (%) or mean (SD)<br>N = 172 | FB n (%) or mean (SD)<br>N = 407 | p |
|---|---|---|---|---|
| **Disability (WHO-DAS ≥20)** | No | 135 (78.5%) | 317 (77.9%) | $X^2 = 0.02$, p = 0.87 |
| | Yes | 37 (21.5%) | 90 (22.1%) | |

EUC enhanced usual care

FB Friendship Bench

PHQ-9 (Patient Health Questionnaire): 0 (no symptoms) to 27 (worst possible symptoms)

GAD-7 (Generalized Anxiety Disorder 7-item Scale): 0 (no symptoms) to 21 (worst possible symptoms)

WHO-DAS (World Health Organization Disability Assessment Schedule): 0 (no difficulty) to 48 (worst possible difficulty)

$ index person sleeps with three or more other persons in the room

* SSQ-14 score ≥11 and either suicidal ideation or hallucinations

were not statistically significant. There was also evidence of an association between FB and symptom severity of common mental disorders (SSQ-14 score) at endline with (aDID = -1.90; 95%CI 3.50 to -0.31; p = 0.02), but no evidence of an association for the other mental health outcomes.

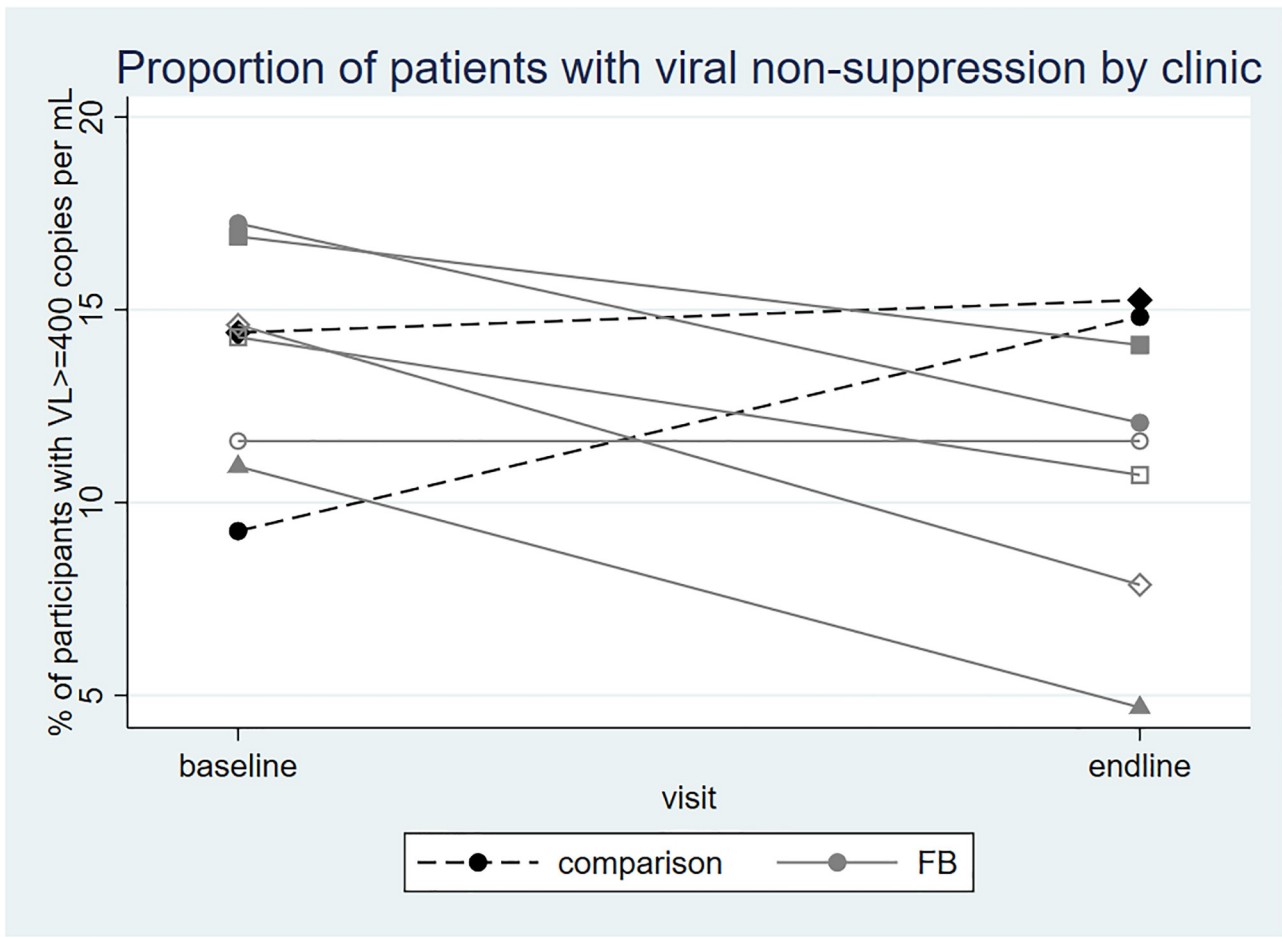

**Fig 2. Proportion of patients with viral non-suppression at baseline and six months follow-up by clinic.**

**Table 2. Viral non-suppression at follow-up by treatment group stratified according to baseline viral load.** Results are difference-in-difference of proportions in viral non-suppression between groups from baseline to follow-up.

| Baseline viral load | Treatment group | N | Follow-up viral load ≥400 copies/ml, n (%) | Difference-in-difference of proportion (95% CI)* | p-value |
|---|---|---|---|---|---|
| NA | EUC | 172 | 26 (15.1%) | -7.3; (-14.7% to -0.01%) | 0.05 |
| NA | FB | 407 | 41 (10.1%) | | |
| <400 copies/ml | EUC | 150 | 14 (9.3%) | -6.5% (-10.5% to -2.4%) | 0.002 |
| | FB | 349 | 10 (2.9%) | | |
| ≥400 copies/ml | EUC | 22 | 12 (54.5%) | -1.1% (-26.3% to 24.1%) | 0.93 |
| | FB | 58 | 31 (53.5%) | | |

EUC: Enhanced usual care

FB: Friendship Bench

*Difference-in-difference of proportions <0 means patients receiving the FB are less likely to have viral non-suppression at follow-up

## Sensitivity analyses

The results for the primary outcome remained robust in both sensitivity analyses. When the sample was restricted to those followed up within 8 months after enrolment (N = 393) the aDiD was -8.1% (95%CI -15.5% to -0.7%; p = 0.04). Similarly, when it was restricted to those enrolled before May 2018 (N = 497), the aDiD was -7.8% (95%CI -16.8% to 1.1%; p = 0.08).

**Table 3. Association of treatment group with mental health using cluster-level analysis adjusted for the respective baseline value.** Results are adjusted DiD of proportions (binary outcomes) or in means (continuous outcomes) between treatment groups from baseline to follow-up.

| | EUC % (n/N) | FB % (n/N) | Difference-in-difference of proportions (95% CI)* | p-value |
|---|---|---|---|---|
| Presence of common mental disorders (SSQ-14 cut-off ≥ 9) | 52.5% (85/162) | 27.6% (112/406) | -21.5% (-36.5% to -6.7%) | 0.008 |
| Presence of depression (PHQ-9 cut-off ≥ 11) | 31.5% (51/162) | 20.9% (85/406) | -0.81% (-28.7% to 12.6%) | 0.41 |
| Presence of anxiety (GAD-7 cut-off ≥ 10) | 28.4% (46/162) | 17.2% (70/406) | -7.7% (-31.0% to 15.6%) | 0.49 |
| Presence of disability (WHO-DAS cut-off ≥ 20) | 14.8% (24/162) | 8.6% (35/406) | -2.9% (-17.7% to 11.9%) | 0.68 |
| | EUC Mean (SD) | FB Mean (SD) | Difference-in-difference in means (95% CI)* | p-value |
| Severity of common mental disorders (SSQ-14 score) | 7.91 (4.17) | 5.55 (3.78) | -1.90 (-3.50 to -0.31) | 0.02 |
| Severity of depression (PHQ-9 score) | 8.20 (5.66) | 6.27 (5.44) | -2.07 (-5.24 to 1.10) | 0.18 |
| Severity of Anxiety (GAD-7 score) | 6.83 (4.83) | 5.31 (4.77) | -1.81 (-4.87 to 1.25) | 0.22 |
| Severity of disability (WHO-DAS score) | 10.85 (8.30) | 7.78 (7.61) | -1.36 (-6.52 to 3.80) | 0.58 |

SSQ-14 (Shona Symptom Questionnaire): 0 (no symptoms) to 14 (worst symptoms)

PHQ-9 (Patient Health Questionnaire): 0 (no symptoms) to 27 (worst possible symptoms)

GAD-7 (Generalized Anxiety Disorder 7-item Scale): 0 (no symptoms) to 21 (worst possible symptoms)

WHO-DAS (World Health Organization Disability Assessment Schedule): 0 (no difficulty) to 48 (worst possible difficulty)

*Difference-in-difference of <0 means patients receiving the FB were less likely to have mental health problems at follow-up

## Discussion

### Main findings

This comparative cohort study in 579 participants accessing ART through primary health care showed that PLWH with co-morbid CMD who received the FB psychological intervention, were more likely to maintain HIV viral suppression than those who received only standard of care mental health treatment based on the WHO mhGAP [23]. Participants who were virally suppressed at baseline were more likely to be virally suppressed at follow-up if they received FB compared to those who received standard of care. However, there was no evidence of a difference in viral non-suppression between mental health treatment groups in participants with viral non-suppression at baseline. This suggests that FB helps PLWH maintain good adherence to ART and stay virally suppressed, but not to newly achieve viral suppression. The burden of common mental disorders in this primary care setting was high, with half the screened PLWH living with clinically relevant depression and/or anxiety.

### Findings of the study in larger research context

To our knowledge, this study is the largest comparative study conducted in sub-Saharan Africa to assess the effect of a brief psychological intervention for CMD among PLWH who are either virally suppressed or virally non-suppressed. The study was implemented during the scale-up of FB through primary care in Zimbabwe, a low-income country with the fifth highest HIV prevalence globally [27].

At the time we conducted this study, there was only preliminary evidence from African countries of the potential to improve viral suppression through treating CMD, based on two pilot studies. One of these conducted in Zimbabwe [28] piloted stepped care for depression integrated with enhanced adherence counselling and one in Cameroon [29] used antidepressants. Both studies found that treatment of depression was associated with increased prevalence of viral suppression, although neither was powered to determine effectiveness [28, 29]. A pre-post study from Malawi compared the FB for mild depression (and antidepressants for moderate to severe depression) with usual care in 501 PLWH newly initiating ART with symptoms of depression (PHQ-9 $\geq$5) [30]. No evidence was found that treatment of depression improved viral suppression as compared to usual care. A possible explanation for this lack of effect may be that interventions for depression were poorly implemented, there was no additional adherence counselling in the intervention arm and loss to follow-up was high. A further reason could be that entry criteria for depression were too low. In the Malawi study only 25% of participants had a PHQ-9 score of 10 or above, compared to 63% in our study. Furthermore, the study was restricted to those newly initiating ART, who may not yet be experiencing treatment fatigue [31]. A recently published cluster-randomised trial from Uganda found that eight sessions of group support psychotherapy, compared to group HIV education, improved depression and self-reported ART adherence at 24 months in PLWH with major depression [32]. This is an encouraging finding but it should be noted that the trial's adherence measure was based on only one self-report question, baseline adherence to ART was very high, and outcome assessors were unblinded, which limits confidence in the adherence outcome. Viral suppression was greater (96%) in the intervention arm, than the control arm (88%) but analysis was not stratified by baseline viral suppression.

Our findings that PLWH with viral non-suppression and CMD did not achieve viral suppression from CMD treatment alone are in keeping with the evidence mainly from the United States that only interventions with a specific adherence-related component resulted in improved adherence [33]. This has been demonstrated recently in South Africa, where nurse-

delivered cognitive behavioural therapy for depression and adherence was effective in improving viral suppression as well as clinical depression for virally unsuppressed PLWH [34]. PLWH who have depression and poor adherence may need adherence-specific interventions to change their behaviour and re-attain viral suppression.

## Strengths and limitations

We assessed CMD using three psychometric tests that were validated in the study setting [19, 21]. The PHQ-9 and the GAD-7 are standard tests to measure depression and anxiety in primary care populations. Trained lay counsellors delivered FB in six primary health care centres in Harare. The effect of FB on CMD and depression is already well documented, and has led to considerable scale-up of the programme in Zimbabwe and other countries. This makes it an ideal intervention to test for its potential effect on HIV viral suppression in PLWH.

A major limitation of the present study is the non-randomised design and consequently the risk of bias due to confounding. Participants at the FB clinics were older, more likely to be female, more likely to be widowed, less likely to have an income, and had been on ART for a longer period. To decrease this risk we evaluated a comprehensive set of baseline characteristics as potential confounders. We only found age and duration since ART initiation to be independently associated with both treatment group and viral non-suppression at endline. These variables did not change the effect estimate of the intervention on the primary outcome by more than 10% suggesting little confounding.

Another limitation is that enrolment was lower than the planned sample size. Time and resource constraints led us to close the study after including 500 participants in 6 clinics instead of 750 participants in 10 clinics in the FB group, and 200 participants in 2 clinics instead of 250 in 3 clinics in the EUC group. The smaller than planned sample size might be an explanation for the lack of statistically significant association between treatment group and depression or anxiety. All associations between treatment group and mental health were in the same direction, suggesting fewer mental health symptoms in the FB group.

Viral non-suppression was less prevalent than anticipated which limited our power to detect a difference between groups at follow-up. Baseline prevalence of viral non-suppression was 13.8%, which is similar to the 14.5% non-suppression found in a population-based survey conducted in Zimbabwe [25]. We had expected that a sample with CMD symptoms would have higher baseline viral non-suppression but this was not the case.

Other limitations include the difference in follow-up time between treatment groups and the lack of data on eligibility during the last three months of recruitment. These differences in follow-up duration occurred because of logistic challenges to data collection. The median follow-up was two months longer in the EUC group than the FB group, giving the EUC group a longer period at risk in which to develop viral non-suppression. This might have led to an underestimation of the effect of treatment on viral non-suppression. In the sensitivity analysis restricted to participants who were followed up within 8 months after enrolment, we found a slight increase in the reduction of the proportion of participants at FB clinics with viral non-suppression as compared to the EUC clinics (-7.3% in the main analysis and -8.1% in the subgroup analysis). The original study design specified 6 months' follow-up. However, only 7 (4.1%) EUC participants were followed up within 6 months, so a restriction at 8 months was used to permit a comparison between groups controlling to some extent for follow-up time.

## Implications for research and public health

We found evidence that the FB helped PLWH to maintain viral suppression, and to have better mental health. It is vitally important to improve integration of mental health care into primary

care for people living with HIV in countries with high HIV burden [35]. More research is needed to understand how to improve viral suppression in PLWH with co-morbid CMD. For instance, using problem-solving therapy to help address barriers to adherence to ART, as well as to tackle depression could improve both mental health and viral suppression in PLWH who are virally unsuppressed. An RCT is currently underway to test this question, in Zimbabwe [36]. This is of public health importance, as viral suppression plays a crucial role in ending the HIV pandemic. Additionally, research is needed to understand which types of CMD in PLWH are most likely to respond to the FB intervention. Qualitative research should be conducted, to deepen the understanding of adherence behaviour in those living with HIV who have CMD, and to explore understand the needs of counsellors in supporting people with adherence difficulties [3].

## Conclusion

People living with co-morbid HIV and common mental disorders can benefit from a low-cost mental health intervention to improve mental health outcomes and maintain viral suppression.

## Supporting information

**S1 Table. Associations between baseline characteristics and treatment group (exposure) or viral non-suppression at endline (outcome) presented as ORs with corresponding p-values from individual-level logistic regression analyses.**
(DOCX)

**S2 Table. Associations between baseline characteristics and completeness of viral load (VL) data presented as ORs with corresponding p-values from individual-level logistic regression analyses.**
(DOCX)

**S1 Checklist. Inclusivity in global research.**
(DOCX)

## Acknowledgments

We would like to acknowledge the City of Harare clinics staff to support the study with patient recruitment, giving access to clinical data records and supporting the collection of blood samples to measure viral load. We also would like to thank the research assistants Nyaradzo Goba, Tichaona Gumunyu, Thembile Gola, and Portia Chiuyu for their support in data collection and coordination of follow-up visits of the patients.

## Author Contributions

**Conceptualization:** Victoria Simms, Melanie A. Abas, Helen A. Weiss, Dixon Chibanda.

**Formal analysis:** Victoria Simms, Helen A. Weiss.

**Funding acquisition:** Melanie A. Abas, Helen A. Weiss, Dixon Chibanda.

**Investigation:** Victoria Simms, Melanie A. Abas, Monika Müller, Epiphania Munetsi, Lloyd Dzapasi, Helen A. Weiss, Dixon Chibanda.

**Methodology:** Melanie A. Abas, Epiphania Munetsi, Lloyd Dzapasi, Helen A. Weiss, Dixon Chibanda.

**Project administration:** Epiphania Munetsi, Lloyd Dzapasi.

**Resources:** Epiphania Munetsi, Lloyd Dzapasi, Dixon Chibanda.

**Supervision:** Dixon Chibanda.

**Writing – original draft:** Victoria Simms, Melanie A. Abas, Monika Müller.

**Writing – review & editing:** Victoria Simms, Melanie A. Abas, Epiphania Munetsi, Lloyd Dzapasi, Helen A. Weiss, Dixon Chibanda.

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
