## [Editor Report · Decision Letter 0]

3 Jan 2023

PGPH-D-22-02032

Effect of a brief psychological intervention for common mental disorders on HIV viral suppression: a non-randomised controlled study of the Friendship Bench in Zimbabwe

Dear Dr. Abas,

Thank you for submitting your manuscript to PLOS Global Public Health. After careful consideration, we feel that it has merit but does not fully meet PLOS Global Public Health’s publication criteria as it currently stands. Therefore, we invite you to submit a revised version of the manuscript that addresses the points raised during the review process.

EDITOR: Please insert comments here and delete this placeholder text when finished. Be sure to:

Indicate which changes you require for acceptance versus which changes you recommendAddress any conflicts between the reviews so that it's clear which advice the authors should followProvide specific feedback from your evaluation of the manuscript

Please ensure that your decision is justified on PLOS Global Public Health’s publication criteria and not, for example, on novelty or perceived impact.

We look forward to receiving your revised manuscript.

Kind regards,

Nilanjana Ghosh

Academic Editor

Journal Requirements:

a. State what role the funders took in the study. If the funders had no role in your study, please state: “The funders had no role in study design, data collection and analysis, decision to publish, or preparation of the manuscript.”

b. If any authors received a salary from any of your funders, please state which authors and which funders.

3. We ask that a manuscript source file is provided at Revision. Please upload your manuscript file as a .doc, .docx, .rtf or .tex.

4. Please provide separate figure files in .tif or .eps format.

5. We noticed that you used “data not shown”/"unpublished data" in the manuscript. We do not allow these references, as the PLOS data access policy requires that all data be either published with the manuscript or made available in a publicly accessible database. Please amend the supplementary material to include the referenced data or remove the references.

6. In the online submission form, you indicated that your data will be submitted to a repository upon acceptance.  We strongly recommend all authors deposit their data before acceptance, as the process can be lengthy and hold up publication timelines. Please note that, though access restrictions are acceptable now, your entire data will need to be made freely accessible if your manuscript is accepted for publication. This policy applies to all data except where public deposition would breach compliance with the protocol approved by your research ethics board. If you are unable to adhere to our open data policy, please kindly revise your statement to explain your reasoning and we will seek the editor's input on an exemption. Please be assured that, once you have provided your new statement, the assessment of your exemption will not hold up the peer review process.

Additional Editor Comments (if provided):

Nicely penned

Background may be written with more focus on rationale of doing the study

Impact of covid on immunisupression n mental health may be added in discussion
---

## [Decision Letter · Decision Letter 1]

6 Jun 2023

PGPH-D-22-02032R1

Effect of a brief psychological intervention for common mental disorders on HIV viral suppression: a non-randomised controlled study of the Friendship Bench in Zimbabwe

Dear Dr. Abas,

Thank you for submitting your manuscript to PLOS Global Public Health. After careful consideration, we feel that it has merit but does not fully meet PLOS Global Public Health’s publication criteria as it currently stands. Therefore, we invite you to submit a revised version of the manuscript that addresses the points raised during the review process.

We look forward to receiving your revised manuscript.

Kind regards,

Claudia P. Cortes, MD

Academic Editor

Journal Requirements:

2. Our staff editors have determined that your manuscript is likely within the scope of our Global Mental Health: challenges, opportunities, and the future of the field. This editorial initiative is headed by a team of Guest Editors for PLOS GPH: Rochelle Burgess (University College of London) and Dixon Chibanda (University of Zimbabwe and London School of Tropical Medicine and Hygiene). The Collection invites researchers to submit original research which engages with, or disrupts, the urgent needs across the global mental health landscape. We especially encourage submissions of studies that critically interrogate the status quo of the field and that involve inter-/trans-disciplinary approaches and those which share perspectives from underrepresented global regions and communities.

 Additional information can be found on our announcement page: https://collections.plos.org/call-for-papers/global-mental-health-opportunities-challenges/ 

If you would like your manuscript to be considered for this collection, please let us know in your cover letter and we will ensure that your paper is treated as if you were responding to this call.  Please note that being considered for the Collection does not require additional peer review beyond the journal’s standard process and will not delay the publication of your manuscript if it is accepted by PLOS GPH. If you would prefer to remove your manuscript from collection consideration, please specify this in the cover letter.

3. We ask that a manuscript source file is provided at Revision. Please upload your manuscript file as a .doc, .docx, .rtf or .tex.

4. We noticed that you used "data not shown" in the manuscript. We do not allow these references, as the PLOS data access policy requires that all data be either published with the manuscript or made available in a publicly accessible database. Please amend the supplementary material to include the referenced data or remove the references.

Additional Editor Comments (if provided):

upon further review it seems to me that the comments of reviewer number 1 are appropriate and should be addressed by the authors.

in any case they try to clarify the text and thus make the reading more friendly and understandable.

Reviewers' comments:

Reviewer's Responses to Questions

**Comments to the Author**

1. If the authors have adequately addressed your comments raised in a previous round of review and you feel that this manuscript is now acceptable for publication, you may indicate that here to bypass the “Comments to the Author” section, enter your conflict of interest statement in the “Confidential to Editor” section, and submit your "Accept" recommendation.

Reviewer #1: (No Response)

Reviewer #2: All comments have been addressed

2. Does this manuscript meet PLOS Global Public Health’s publication criteria? Is the manuscript technically sound, and do the data support the conclusions? The manuscript must describe methodologically and ethically rigorous research with conclusions that are appropriately drawn based on the data presented.

Reviewer #1: Yes

Reviewer #2: Yes

3. Has the statistical analysis been performed appropriately and rigorously?

Reviewer #1: Yes

Reviewer #2: Yes

4. Have the authors made all data underlying the findings in their manuscript fully available (please refer to the Data Availability Statement at the start of the manuscript PDF file)?

Reviewer #1: Yes

Reviewer #2: Yes

5. Is the manuscript presented in an intelligible fashion and written in standard English?

Reviewer #1: Yes

Reviewer #2: Yes

6. Review Comments to the Author

Reviewer #1: Review: Effect of a brief psychological intervention for common mental disorders on HIV viral

suppression: a non-randomised controlled study of the Friendship Bench in Zimbabwe.

Abstract

The abstract written well.

Introduction

Although the rationale for conducting the study is well explained and the exposure in the study appears to be FB on individuals with CMD, the hypothesis leaves out the CMD aspect. Shouldn’t it read, “individuals with CMD receiving FB intervention compared to the same receiving EUC”? The authors need to be clear on this.

Materials and methods

Study design and setting

The study design and setting are well described.

Recruitment

The procedure of recruiting participants is well described. However, it is unclear what training the supervisor had and what tools he/she used to screen for psychosis, suicidality etc? Is it the same supervisor who screened those participants from FB and the EUC?

Data collection baseline and endline

Data collection tools are well described. However, the authors need to be clear about the length of time the participants received the FB intervention before enrolling as the intervention has already been shown to reduce the mental health symptoms. As the authors used a cut off less than 400 to determine whether one has a positive or negative outcome, it would be helpful to know the levels of suppression using the log viral load in the two groups and whether there were any differences between the groups. If the FB group had already received help from the intervention, the absolute viral loads would already be low and any further exposure to the FB would improve it.

Intervention

FB

This is well described.

EUC

This is well described.

HIV Care

This is well described.

Outcomes

These are well described and use validated tools as needed.

Statistical analysis

This is well described.

Results

Study flow

This is well described.

Participants characteristics

This section is presented very well.

Association of FB with viral non-suppression

In the absence of data as to the duration between the entry into the study and the duration this group has been receiving the FB psychological intervention, it is not possible to draw the conclusion on the association of the FB and viral non-suppression. It is also important to have a numerical measure of viral load e.g., log viral load etc so that these can be compared between the FB and the EUC.

Association of FB with mental health

Information needs to be provided on the duration of FB intervention before the participants were recruited into the study. The participants were already getting the benefit of being in the FB intervention.

Sensitivity analysis

Discussion

The discussion is beautifully written.

Strength and limitations

The other limitation of note is the use of viral loads as a proportion, and use of a relatively high cut-off of 400. <40 is usually considered as virally suppressed.

Conclusion

Given the findings, the conclusion is proper.

Reviewer #2: A very interesting work regarding the role of social support on treatment.

Some minor comments regarding language and redaction.

7. PLOS authors have the option to publish the peer review history of their article (what does this mean?). If published, this will include your full peer review and any attached files.

**Do you want your identity to be public for this peer review?** For information about this choice, including consent withdrawal, please see our Privacy Policy.

Reviewer #1: No

Reviewer #2: No

---

## [Decision Letter · Decision Letter 2]

3 Oct 2023

Effect of a brief psychological intervention for common mental disorders on HIV viral suppression: a non-randomised controlled study of the Friendship Bench in Zimbabwe

PGPH-D-22-02032R2

Dear Professor Abas,

We are pleased to inform you that your manuscript 'Effect of a brief psychological intervention for common mental disorders on HIV viral suppression: a non-randomised controlled study of the Friendship Bench in Zimbabwe' has been provisionally accepted for publication in PLOS Global Public Health.

Best regards,

Siyan Yi, MD, MHSc, PhD

Academic Editor

Reviewer Comments (if any, and for reference):

Reviewer's Responses to Questions

**Comments to the Author**

1. If the authors have adequately addressed your comments raised in a previous round of review and you feel that this manuscript is now acceptable for publication, you may indicate that here to bypass the “Comments to the Author” section, enter your conflict of interest statement in the “Confidential to Editor” section, and submit your "Accept" recommendation.

Reviewer #1: All comments have been addressed

2. Does this manuscript meet PLOS Global Public Health’s publication criteria? Is the manuscript technically sound, and do the data support the conclusions? The manuscript must describe methodologically and ethically rigorous research with conclusions that are appropriately drawn based on the data presented.

Reviewer #1: Yes

3. Has the statistical analysis been performed appropriately and rigorously?

Reviewer #1: Yes

4. Have the authors made all data underlying the findings in their manuscript fully available (please refer to the Data Availability Statement at the start of the manuscript PDF file)?

Reviewer #1: Yes

5. Is the manuscript presented in an intelligible fashion and written in standard English?

Reviewer #1: Yes

6. Review Comments to the Author

Reviewer #1: Dear Editor

i appreciate the revisions by the authors and I am happy to recommend the manuscript for publication.

Well done authors.

7. PLOS authors have the option to publish the peer review history of their article (what does this mean?). If published, this will include your full peer review and any attached files.

**Do you want your identity to be public for this peer review?** For information about this choice, including consent withdrawal, please see our Privacy Policy.

Reviewer #1: **Yes: **Dr Munyaradzi Madhombiro
